# Cracking double-blind review: Authorship attribution with deep learning

**Leonard Bauersfeld**[ID]<sup>☯</sup>*, **Angel Romero**[ID]<sup>☯</sup>, **Manasi Muglikar**, **Davide Scaramuzza**

Robotics and Perception Group, University of Zurich, Zurich, Switzerland

☯ These authors contributed equally to this work.
* bauersfeld@ifi.uzh.ch

**Data Availability Statement:** The data underlying the results presented in the study are available from (https://arxiv.org/).

## Abstract

Double-blind peer review is considered a pillar of academic research because it is perceived to ensure a fair, unbiased, and fact-centered scientific discussion. Yet, experienced researchers can often correctly guess from which research group an anonymous submission originates, biasing the peer-review process. In this work, we present a transformer-based, neural-network architecture that only uses the text content and the author names in the bibliography to attribute an anonymous manuscript to an author. To train and evaluate our method, we created the largest authorship-identification dataset to date. It leverages all research papers publicly available on arXiv amounting to over 2 million manuscripts. In arXiv-subsets with up to 2,000 different authors, our method achieves an unprecedented authorship attribution accuracy, where up to 73% of papers are attributed correctly. We present a scaling analysis to highlight the applicability of the proposed method to even larger datasets when sufficient compute capabilities are more widely available to the academic community. Furthermore, we analyze the attribution accuracy in settings where the goal is to identify all authors of an anonymous manuscript. Thanks to our method, we are not only able to predict the author of an anonymous work but we also provide empirical evidence of the key aspects that make a paper attributable. We have open-sourced the necessary tools to reproduce our experiments.

## 1 Introduction

Most known academic and literary texts can easily be attributed to a certain author because they are signed. Yet sometimes, we find anonymous pieces of work and would like to identify an author based on the given text, a method referred to as author attribution (AA).

The AA problem is particularly interesting in the context of double-blind peer review in academic research, a technique often implemented to robustify the process against human biases. By addressing the AA task for research papers, we aim to not only demonstrate the technical feasibility of large-scale authorship attribution but hope to improve the double-blind peer review process by providing empirical evidence of the key aspects of a paper that allow experienced reviewers to correctly guess which group of authors a certain manuscript originated from. Especially for research papers, AA is a complex task due to the vast number of possible authors, the length of the texts, and the unavailability of a large-scale dataset.

**Funding:** This work was supported by the National Centre of Competence in Research (NCCR) Robotics through the Swiss National Science Foundation (SNSF) and the European Research Council (ERC) under grant agreement No. 864042 (AGILEFLIGHT). The funders had no role in study design, data collection and analysis, decision to publish, or preparation of the manuscript.

**Competing interests:** The authors have declared that no competing interests exist.

Author attribution for literary texts first became popular in 1964 when researchers studied the famous "The Federalist" papers [1], a collection of 85 articles and essays published under the pseudonym "Publius", to identify the authors who contributed to each essay. More recently, authorship attribution for rulings written by the Australian High Court [2] and internet blogs [3] has been studied. Scientific texts, however, are inherently different from the aforementioned works as individual authors are not only identifiable by a certain writing style but most likely write on similar topics in their works and cite themselves more often. Furthermore, no large-scale authorship attribution dataset for academic texts exists.

We aim to address both challenges: this work presents a novel architecture (summarized in Fig 1) alongside a new dataset to address the problem of AA for research papers. Instead of just using the text content [4], our method relies on both text content and the author names of the paper cited in the *Reference* section of a manuscript, discarding all image data and equations. Following the latest advances in natural language processing, the transformer Distil-BERT [5] is used to process the text section. For the references, a frequency histogram-embedding with a subsequent multi-layer perceptron is used. We leverage all publicly available arXiv [6] submissions, that amount to more than 2 million research papers, to construct a new dataset tailored to this hybrid AA approach. The dataset includes text content as well as the references cited in a paper. On the largest arXiv-subset with 2070 candidate authors, we achieve an AA accuracy of 73.4%, while, on smaller sets with 50 possible authors, well over 90%. We find that already the first 512 words of a manuscript (including the abstract, when available, and parts of the introduction) lead to more than 60% of the papers being attributed correctly. Furthermore, the experiments clearly show that self-citations improve attribution accuracy by up to 25% percentage points compared to when self citations are omitted. When the goal is to identify all authors of a manuscript, we still achieve and impressive 50% accuracy for papers with more than two authors, given the number of authors to be identified is known. If the exact number of authors is unknown, but one is only interested in a set of 5 candidate authors that could be authors, in 67% of the cases all candidate authors are among the top-five suggestions of the model.

### 1.1 Contributions

In summary, we make the following contributions. First, we present a novel deep-learning-based architecture capable of analysing and classifying hundreds of thousands of research texts and references from arXiv to address the AA problem. Second, to train this architecture, we build a large-scale dataset based on the research publications available on arXiv. Subsequently, an analysis of the attribution accuracy, and the scalability of our method is presented alongside empirical evidence showing which are the key aspects that make a paper attributable. Possible applications of our work go beyond the double-blind peer review process and we briefly discuss a possible application in plagiarism detection as well as simple steps that can be easily adapted during the review process to anonymize the papers better.

### 2 Related work

Perhaps one of the oldest examples of authorship attribution (AA) was to identify the co-writers of William Shakespeare in 36 plays (collectively called *"Shakespeare canon"* [7]), which began in the late 17th century. The research on authorship attribution became much more popular in 1964 when researchers studied "The Federalist" papers [1]. After this, AA advanced through the development of more involved hand-crafted feature extractors for text, resulting in over 1000 different published approaches by the year 2000 [8, 9]. Subsequently, the computer-assisted approaches were further automated, and prior to the machine learning era, two

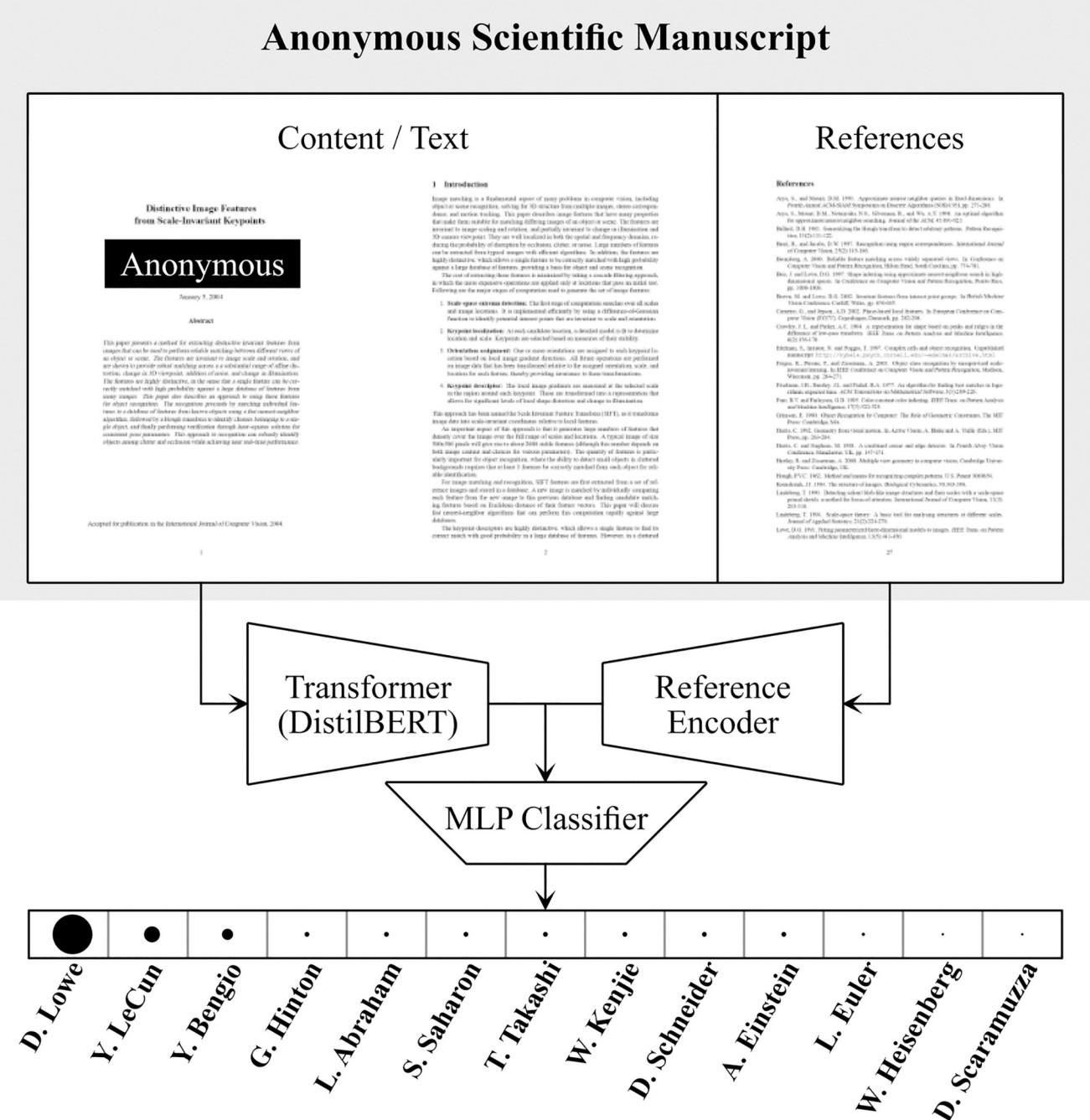

**Fig 1. Our method identifies authors of anonymous scientific manuscripts by leveraging both the information contained in the text as well as the citations.** We encode the main text using DistilBERT [5] and combine this encoding with a feature vector extracted from the cited references. The encodings are subsequently fused by a two-layer classification MLP. It outputs the log-likelyhoods that the given anonymous paper has been (co-)authored by one of the over 2000 authors included in our novel dataset.

dominant approaches existed: profile-based AA and instance-based AA (IAA) [10]. The former extracts one feature vector (author profile) per author and compares the feature distance of a given text with all author profiles, whereas IAA extracts a feature vector per text sample and uses a classifier (e.g. SVM [10]) to distinguish authors.

Along with the rise of short electronic messages (e-mails, tweets) came a growing interest in text classification (e.g., hate speech [11], polarizing rhetoric tweet analysis [12]) and AA using short texts (e.g., detect 'hacked' accounts [13]). Machine learning proved to be vital for this task since learned feature descriptors like *document embedding* [14] outperform classic character n-gram [15] and bag-of-words approaches. N-gram convolutional nets also show competitive performance [16].

From a text-length perspective, research papers are more similar to news articles and books than to tweets. In [17] a uni-gram feature in combination with an SVM is used for news articles and book AA, and they achieve 83% classification accuracy on a dataset with 50 authors. In [18] a study comparing different network architectures (LSTM, GRU, Siamese network) on similar data is presented, and a near-perfect classification is achieved, also on a dataset with only 50 different authors. In [19, 20] it is confirmed that deep networks achieve very competitive performance on AA and authorship profiling (AP) tasks. The results obtained on public benchmark datasets in those works are used as *baselines*, although they are focused on single-author documents (non-research articles) and are only applied to the comparably small benchmarks.

For research papers, solving authorship attribution is a more complex task due to the length of the texts, their heterogeneity (mathematical symbols, reference sections, etc.), and the vast amount of possible authors. Therefore, authorship attribution has been applied to research articles only in very rare cases, such as [4], where the (not publicly available) training and testing datasets are rather limited (403 authors, 1683 papers).

The recent advances in natural language processing (NLP), namely the development of transformer-based architectures, allow us to tackle these difficulties. Transformers have shown impressive capabilities in NLP for mid-sized text lengths, e.g., BERT [21], its smaller counterpart DistilBERT [5], and BigBird, for longer sized sequences [22]. The success of transformers has enabled applications such as ancient text restoration and attribution, polarizing tweet analysis [12], hate speech detection [23], emotion recognition in conversation [24] and song analysis [25]. In [26] results indicate the usefulness of using such networks as feature extractor.

The increasingly large number of studies on the use of scientific documents with bibliometric applications shows the growing interest of the bibliometric community in authorship attribution [27]. Specifically, machine learning applied to bibliometrics has steadily been getting more traction in the last decade [28]. In [29], the authors analyse the use of solely the reference section to predict the possible authors of scholarly papers. However, all the aforementioned research focuses either on the analysis of the texts themselves or solely on the references. To the best of the authors' knowledge, this work presents the first approach, where both sources of information are combined.

## 3 Dataset

In contrast to the works on authorship attribution for news articles, legal documents or blog entries, this work focuses on research articles. Therefore, the benchmark datasets that are commonly used are not suitable, and a new dataset based on arXiv articles is developed. This section first introduces our arXiv dataset, then the standard benchmarks are briefly described, and a brief discussion of the challenges and features is presented.

### 3.1 The arXiv dataset

The arXiv is an open-access preprint server for scientific papers in the field of computer science, math and physics, which contains over 2 million research articles at the time of writing (https://arxiv.org/). The pdf versions of the articles can be downloaded [6] together with a

database file that maps the unique arXiv-identifier (e.g. 2106.08015) to the title of the paper, the authors' names and the abstract. Note that, unfortunately, no UUIDs (unique user identifiers) are assigned to authors on arXiv, which causes ambiguity between different authors with the same name.

**3.1.1 Preprocessing.**   In order to reduce the name ambiguity, a first step discards all entries where the authors did not provide their full names but only initials. Subsequently, all authors with at least $P$ papers are selected to yield a dataset named $D(\$P)$, e.g. for $P = 300$ the dataset is D300. All co-authors are treated equally as not all fields order the authors by the amount of contribution.

For all papers in the dataset, the plain-text version of their articles is loaded and processed. In the given order, this processing

1. discards the header containing the title, the authors' names, contact info and affiliation,

2. extracts the content (abstract and body) of the paper,

3. extracts the 'References' section,

4. splits the reference section into individual references,

5. extracts the cited authors' names from the references.

All splitting and extraction of parts are done using hand-crafted regular expressions that are 'fail-fast', meaning that if they succeeded in segmenting the paper, the result is almost always correct (e.g. a human performing the same task would segment similarly). The processing removes about 30% of the samples from the dataset for all values of $P$.

We now summarize the functionalities implemented to preprocess the documents. For further details we refer the reader to the open-source code to see the exact routines used for preprocessing the document. In a first step, all blank lines, lines containing email addresses and lines containing only a digit are removed. To discard the header, the keyword 'abstract' is used as 75% of the papers contain 'abstract' in their text. The authors, title, and affiliations are given before the abstract and can thus be trimmed. The main text of a paper is assumed to be terminated by the keyword 'references', or '[1]' to indicate the start of the references section (95% of the papers contain this). Many texts have been found to contain a supplement section, supplementary tables, figures or similar further materials after the references. These are identified by the corresponding keyword (e.g. 'supplement', 'appendix', 'discussion'). Such sections are discarded and neither considered part of the main paper content nor the references. A keyword in the above sense means that the word starts with a capital letter and can only be preceded by whitespace in the current line.

Note that even in cases where the preprocessing erroneously would not remove the authors' names (although a manual check did not reveal a single instance of this failure mode), this has no influence on the author attribution as almost no names are part of the vocabulary used by the DistilBERT.

**3.1.2 References.**   The preprocessing of the text yields two text blocks: the main content and the references. Since the proposed method leverages the author names of the cited papers, they need to be extracted from the cited references. The underlying idea of the implementation is based on the assumption that most bibliographies follow one of the standards APA, MLA, Chicago, Angewante Chemie, IEEE or ACL and we can thus extract the cited authors' last names directly from the manuscript. While this approach has limitations as it ignores authors names covered by the 'et al.' statement and does not disambiguiate between different authors with the same last name, we find it a very effective way to extract information from the references given the limited information available on arXiv manuscripts.

Approaches such as matching against an database would require a low-level API access to check hundreds of thousands of papers. Furthermore, it can be very difficult as some citations styles (e.g. Angewandte Chemie) do not list the paper title in the references section, but only (ambiguous) author names and years which severely complicates matching against a database.

In the remainder of this section, an overview over the authors' surname extraction approach is given, and for further details on the exact regular expressions used, the reader is referred to the open-source code. For most styles the individual references can be separated, for example, by splitting on dots '.' at the end of a line or a '[x]' at the beginning of the line. As all cited references are written in the same style, the splitting mechanism is determined by analyzing all references at once and then selecting the appropriate separators. This selection is done automatically and checks different separators until one that is considered successful (i.e. enough splits are found and references have a plausible length).

After the individual references are split, the names of the cited authors need to be extracted. Once again, we note that all commonly used citation styles start a reference with the names of the authors and therefore the list of names can be extracted by taking the start of each reference and trim it once quotes '"", brackets '(' or '[' are encountered. The last step is to identify the delimiter used to separate the names of the authors and split accordingly and only the last part of each name (presumably the surname) is kept. The delimiter is identified as the most-frequent non-alphanumeric, non-whitespace character in the section containing the names. The pre-processing methodology was developed incrementally by manually checking the failure cases and addressing them, which leads to the rule-based approach being robust and performing similar to a human curating the data.

After this processing step, for each paper, we obtain a list of cited papers and for each of these cited papers, the last names of all mentioned authors are listed.

**3.1.3 Author ambiguity.**   Because arXiv lacks UUIDs, the authors are only identified by their full names. This ambiguity became especially obvious for short names which had over 10000 papers assigned to them. To resolve this issue, a clustering approach is used: using a pretrained sentence transformer [30] to extract a feature embedding from the abstract. Then, DBSCAN is used to cluster the extracted feature vectors. If DBSCAN finds only one cluster and some noise, the author is assumed to be one physical person, whereas multiple clusters are identified for ambiguous authors. DBSCAN has been tuned to correctly classify a set of 20 known, unique authors (famous researchers with distinctive names) and 20 known ambiguous authors (checked via Google Scholar).

For datasets with a high threshold $P$, over half of the authors are discarded, whereas only 20-25% of the authors are removed for lower thresholds. This follows the intuition that no single physical person will have published 5000 papers, but 100 is certainly possible through co-authorships.

Note that because of the preprocessing, papers get discarded, and thus there are authors with fewer papers than the original threshold. The D100 dataset has, for example, on average, only 99 papers per author and as few as 25 manuscripts per author in many cases.

**3.1.4 Content chunks.**   Transformer architectures scale badly with the sequence length, which is why most networks have a hard limit between 256 and 4096 tokens. The DistilBERT network can process up to 512 tokens per text. Therefore, the content of the paper is divided into multiple chunks of length up to 512 words. Either the first chunk only (referred to as Dxxx, e.g. D300) or all chunks are used (referred to as Dxxx-C, e.g. D300-C). The rationale for using the first 512 tokens is that those contain the abstract and introduction, which usually summarize the whole paper. While the first 512 words of a paper almost never contain

equations or tables, later chunks can. In tables and equations, the individual symbols are always surrounded by white space. Therefore, all chunks that have an average word length below 4.22 characters are discarded, as they are assumed to primarily consist of tables and equations. This threshold is computed as the $5^{th}$ percentile of a distribution of the average word length in a 512 word (general English) text. The individual word lengths in this text follow the distribution of word lengths in English texts [31].

In a final step, an 80/20 division into train/test dataset is performed. This random split is done using stratified sampling, such that the 80/20 balance is kept for each author. If a paper in the training set was authored by multiple authors in the dataset, it is randomly assigned to one of them. Papers in the test set can contain multiple authors and are correctly classified if the network predicts that it was written by *one* of its authors. Furthermore, it is ensured that a co-authored paper can not be in the train and test split at the same time. An overview of the different arXiv datasets is given in Table 1.

**3.1.5 Trimmed datasets.** The datasets presented above always encompass as many papers per author as are available in arXiv. To better understand the influence of the amount of papers per author on the accuracy of the prediction, we also use a series of trimmed datasets where the 80/20 train/test split is maintained, but the overall number of papers per author is artificially limited. To denote a dataset where only xx randomly sampled papers are used, we append Txx. For example, a D200-C dataset were we limited ourselves to 25 papers per author would be called D200T25-C. Similar to the original D200-C this dataset still contains 226 authors but now with only 25 texts per author instead of 213 texts per author on average. To ablate how the amount of authors influences the accuracy of the authorship attribution a set of trimmed datasets is generated based on the D200-C with 200, 100, 50, and 25 papers per author.

Furthermore, using trimmed datasets enables training with even more authors and a trimmed version of the D50T25-C dataset with only 25 authors per paper but over 5000 candidate authors is generated. Therefore, trimmed datasets enable us to evaluate the scalability of our approach with an increasing number of possible authors while partially avoiding the computational burden associated with larger datasets.

**Table 1. Summary of the datasets used in this work.**

| Dataset | | Authors | Words | Words/Text | Texts/Aut. | Words/Aut. |
|---|---|---|---|---|---|---|
| Benchmarks | Legal | 3 | 3.1M | 2312 | 447 | 1.03 M |
| | Blog10 | 10 | 2.1M | 91 | 2305 | 212 k |
| | Blog50 | 50 | 7.2M | 98 | 1466 | 144 k |
| | Reuters50 | 50 | 2.5M | 506 | 100 | 50 k |
| | IMDb62 | 62 | 21.7 M | 349 | 1000 | 350 k |
| Our arXiv Dataset | D500 | 7 | 1.7 M | 512 | 472 | 241 k |
| | D500-C | 7 | 17.1 M | 5184 | 472 | 2.4 M |
| | D400 | 13 | 2.8 M | 512 | 425 | 217 k |
| | D400-C | 13 | 31.3 M | 5658 | 425 | 2.4 M |
| | D300 | 49 | 8.0 M | 512 | 320 | 164 k |
| | D300-C | 49 | 94.3 M | 6024 | 320 | 2.1 M |
| | D200 | 226 | 24.6 M | 512 | 213 | 109 k |
| | D200-C | 226 | 289 M | 6016 | 213 | 1.2 M |
| | D100 | 2070 | 105 M | 512 | 99 | 50 k |
| | D100-C | 2070 | 1.27 B | 6180 | 99 | 0.6 M |

## 3.2 Benchmark datasets

**3.2.1 Legal.** This dataset consists of written rulings by three Australian High-Court judges from the year 1913 to 1975. Originally, this dataset was used to show that Judge Dixon was ghostwriting for the other two [2]. However, by only using the time period where ghostwriting was impossible, a clean dataset with long texts can be obtained, which is used as a benchmark [20, 32].

**3.2.2 Blog10 and Blog50.** The Blog dataset consists of online blog posts from the years 2002 to 2004 [33]. Most of the posts are very short and often contain rather explicit language. The Blog10 and Blog50 datasets include posts from the top 10 or 50 authors, respectively when sorted by the number of posts [3].

**3.2.3 Reuters50.** The Reuters50 (or CCAT50) is the most widely used [18, 20, 34] AA dataset. It contains news stories and is an excerpt from the Reuters Corpus Volume 1 [35]. The top 50 authors (according to the number of stories) have been selected, and for each author, 100 texts are provided, equally split into a training and a test set [34].

**3.2.4 IMDb62.** The IMDb62 dataset [36] consists of movie reviews from the most active 62 IMDb users, where 1000 texts are provided per author. It is also a very common dataset for benchmarking [3, 10, 20].

## 3.3 Discussion

Compared to the benchmarks, our dataset contains significantly longer texts per author, although there are fewer texts on average per author. Especially the big arXiv datasets (e.g. D100-C) are extremely different than benchmarks like *Blog10* or the *Reuters50*. For example, D100-C contains 600 times more data than *Blog10*. Only the *Legal* and the *IMDb62* datasets are somewhat similar to the small arXiv D400 and D500 in terms of text length and dataset size.

The main difference between the existing AA datasets and the arXiv dataset is that the latter includes an additional feature: the author names of the cited papers. Exploiting this additional information specific to scientific articles is a key contribution of our work. For research article AA no benchmarks exist.

## 4 Architecture

In this section, we present the architecture and sub-architectures (see Fig 2) that are used throughout this paper.

## 4.1 DistilBERT

First we present the architecture that has been used to process the main text of the papers (without the references). For this task, we have chosen to use DistilBERT, a transformer architecture based on a distilled version of BERT. It is smaller, faster, cheaper, and lighter, offering up to 60% faster speeds than BERT while retaining 97% of its language understanding capabilities [5]. In order to convert the raw text to a format that the DistilBERT architecture can take as input, a tokenizer to convert the words to tokens needs to be used. The tokenizer used in our case is based on WordPiece [37]. Both the DistilBERT transformer model and the tokenizer have been initialized with a pre-trained version. Specifically, we use the checkpoint called *distilbert-base-uncased*, which was pre-trained on BookCorpus [38] and the English Wikipedia.

The transformer architecture (*DistilBERT* in Fig 2) consists of an input embedding layer, where the input tokens are mapped into a sequence of vectors, followed by 6 transformer

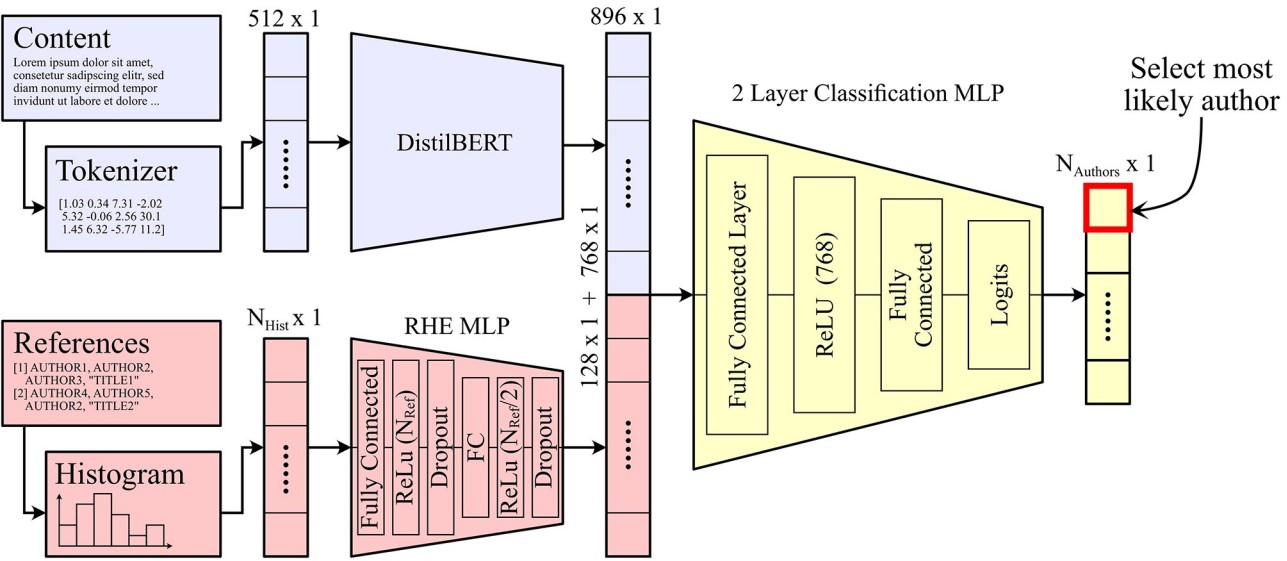

**Fig 2. Our proposed network architecture consists of two separate feature encoders for the different input modalities followed by an MLP network with a logit output layer.**

blocks and a dense output layer. Each transformer block has two sub-layers: a self-attention layer and a feedforward layer. The self-attention layer allows the model to attend to different parts of the input text, while the feedforward layer applies a non-linear transformation to the output of the self-attention layer. The output of the final transformer layer is pooled to produce a fixed-size vector representation of the input text. In our case, the output vector of the final transformer layer (of size 768) is then concatenated to the output layer of the Reference Histogram Embeddings (introduced in the next section) and used as input for the downstream classification task.

One of the main limitations of most transformer architectures is that they have a limited input size. In the case of DistilBERT, this limit is 512 tokens which means that either only the first 512 tokens can be used or the dataset is divided into chunks as described in the previous section. One solution that has been tried to solve this problem is to use BigBird [22], which is a transformer architecture specifically designed for longer sequences and that offers an input limit of 4096 tokens. However, due to the increased size and complexity of the BigBird transformer, training times are very long, and there is no noticeable increase in accuracy compared to a chunked dataset trained with DistilBERT. When training on such chunked datasets, the chunks are processed independently of each other. During the evaluation, all parts of the text that come from the same paper are evaluated consecutively, and the output logits are averaged before converting them to probabilities and selecting the author with the highest probability. As we will show in the results part, this approach proves to be extremely successful and improves the evaluation accuracy by 5-10% (absolute) when compared to only inputting the first part of the paper to DistilBERT.

## 4.2 Reference histogram embeddings

There are many different ways of extracting the key information from the references section of a paper. One of the most direct ones, and the one that resembles the most of how a human reader would do it, is looking directly at the relative frequency of appearance of different

author names. To do this, all the extracted author names (see Section 3) in the dataset are concatenated to build a vocabulary. Only authors that appear frequently (more than 50 times) are added to the vocabulary of size $N_{\text{Hist}}$. Next, for every paper, we create a vector with the same number of elements as the vocabulary, which contains the number of times that each author in the vocabulary appears in the reference section of that paper. This vector is what we call the Reference Histogram Embedding (RHE).

Once we have the RHE for each paper, it is passed through a 2 layer MLP that compresses it. The input layer of this MLP is of size $N_{\text{Hist}}$, the middle layer is of size $(N_{\text{Hist}}+ 128)/2$ and the output layer is of size 128. The output vector is then directly concatenated with the output embeddings of the DistilBERT architecture. This joint vector is then fed to the 2-layer classifier, as shown in Fig 2.

### 4.3 Alternative architecture

We also studied an alternative architecture to include the references in our approach. Since the author names are one of the most informative part of the references, we only encode the author names in the references using FastText [15]. FastText was trained by inputting together all the author names corresponding to one paper, for all papers. This learnt embedding space clusters author names if they are cited by the same paper. Since each paper may have a different number of references and author names, in this architecture we propose to use an LSTM architecture. The input to the LSTM is the variable-length sequence of the author embedding. The output of LSTM concatenated with DistilBERT embedding is passed to the 2-layer MLP.

## 5 Results

This section presents the results achieved using the proposed architecture presented in the previous section and depicted in Fig 2. This architecture has been implemented using the *Hugging Face* library for transformers [39]. First, we compare the performance of our approach to existing approaches on benchmark datasets unrelated to scientific research article AA. Then, we present results on our new *arXiv* dataset are along with a detailed analysis of the prediction accuracy, the scalability to larger datasets and an ablation study of the optimal learning rate. Finally we present a small ablation on the network architecture itself.

### 5.1 Baselines

To evaluate the performance of the network architecture presented in this work, it is compared with current state-of-the-art methods on the benchmark datasets introduced in 3.2. Note that only the 'Content' part using the DistilBERT is used because no benchmark for research articles AA exists. The learning rate of the DistilBERT has been fine-tuned for the datasets and is set to 2e-5 for all experiments. The results are summarized in Table 2.

**Table 2. Comparison of our DistilBERT ("Content") architecture with other methods on the most common authorship attribution benchmark datasets.**

| | Legal | Blog10 | Blog50 | Reuters50 | | IMDb62 |
|---|---|---|---|---|---|---|
| **Train/Test Split** | **80/20** | **80/20** | **80/20** | **50/50** | **90/10** | **80/20** |
| Topic Model [32] | 93.64 | – | – | – | – | 91.79 |
| Article GRU [18] | – | – | – | – | 69.1 | – |
| N-Gram [20] | 91.29 | – | – | **72.6** | – | 94.8 |
| BertAA [3] | – | **65.4** | **59.7** | – | – | 90.7 |
| **DistilBERT (Ours)** | **94.8** | 64.3 | 59.1 | 66.5 | **83.6** | **97.5** |

On the larger *Legal* and *IMDb62* dataset, our DistilBERT approach outperforms all baselines and nearly halves the error rate on *IMBd62*. On the smaller Blog datasets, the transformer-based BertAA [3] approach sightly outperforms ours by about 1%. On the original *Reuters50* dataset, the classical n-gram approach [20] achieves a 6% (absolute) higher accuracy compared to DistilBERT. This is most likely because the transformer-based approach requires much more training data. This theory is supported by the superior results when using a 90/10 train/test split and also by [20], where a similar tendency is observed.

## 5.2 Our dataset

When applying our approach to the arXiv dataset, different network architectures are possible, namely a) only content ("Content"), b) only references with and without self-citations ("References", "Ref (no self)") and c) content with references ("Ref+Cont"). The results for all architectures applied to all versions of the dataset are summarized in Table 3, and visualized in Fig 3.

From Table 3 it is visible that including the references in almost all cases—as expected—increases the accuracy by up to 8%. When all self-citations are removed from the references ("Ref (no self)") the accuracy of a reference-only design drops by over 10 p.p. (percentage points) for the large D100 set. Furthermore, a boost in performance is visible when comparing the non-C datasets (first 512 words only) with the whole documents. This is especially pronounced in the large datasets where relative improvements of up to 30% (content only) are observed. However, this boost in performance comes at the cost of dramatically increased training times, as shown in Table 4.

It is also important to remark that gains in evaluation accuracy that are attributed to solely combining the references with the content come nearly for free in terms of training time, as training times are similar with and without the reference part added to our architecture. The evaluation boost is more prominent when dealing with bigger datasets where only the first 512 are used. For example, it is interesting to see in Table 3 for the column D100 that a) there is an absolute gain of almost 11% when using the references combined with the content w.r.t. only using the content information; and b) the reference alone architecture yields a 54.3% prediction accuracy, a result that is impressive by itself. Even more so, as D100 is a dataset that has more than 2000 possible labels. However, for the chunked datasets, the accuracy increase attributed to the consideration of the references gets diminished, although, for D100-C, D300-C, D400-C and D500-C in Table 3 the best results (by a small margin) are obtained through this strategy. The reason for this decrease in the difference is thought to be related to the nature of transformers. It is known that transformer architectures excel at large amounts of data [3]. This is evident in our experiments when comparing the chunked versions with the first-words-only ones. It is, therefore, to be expected that the increase in performance when adding the reference information is less prominent for an intensively trained transformer.

**Table 3. This table summarizes the authorship identification accuracy in % on the test split of the different arXiv datasets four our method.** On the largest dataset D100-C our approach achieves 73.4% correct authorship attribution.

| Input | Epochs | Only first 512 Words | | | | | | Document in Chunks of 512 Words | | | | | |
| | | LR | D100 | D200 | D300 | D400 | D500 | LR | D100-C | D200-C | D300-C | D400-C | D500-C |
| --- | --- | --- | --- | --- | --- | --- | --- | --- | --- | --- | --- | --- | --- |
| Content | 40 | 1e-5 | 45.5 | 64.7 | 78.9 | 89.0 | 92.0 | 1e-5 | 70.0 | 84.8 | **90.3** | 94.8 | 96.3 |
| Content | 10 | 1e-4 | 49.7 | 68.9 | 82.4 | 91.3 | 92.9 | 1e-4 | | **85.6** | **90.3** | 94.4 | 95.7 |
| References | 10 | 8e-4 | 54.3 | 71.0 | 79.8 | 89.6 | 90.2 | 8e-4 | 54.3 | 71 | 79.8 | 89.6 | 90.2 |
| Ref (no self) | 10 | 8e-4 | 43.3 | 62.9 | 75.5 | 84.3 | 86.4 | 8e-4 | 43.3 | 62.9 | 75.5 | 84.3 | 86.4 |
| **Ref+Cont** | 10 | 3e-4 | **60.5** | **79.0** | **87.0** | **94.3** | **93.1** | 5e-5 | **73.4** | 81.1 | **90.3** | **96.0** | **96.6** |

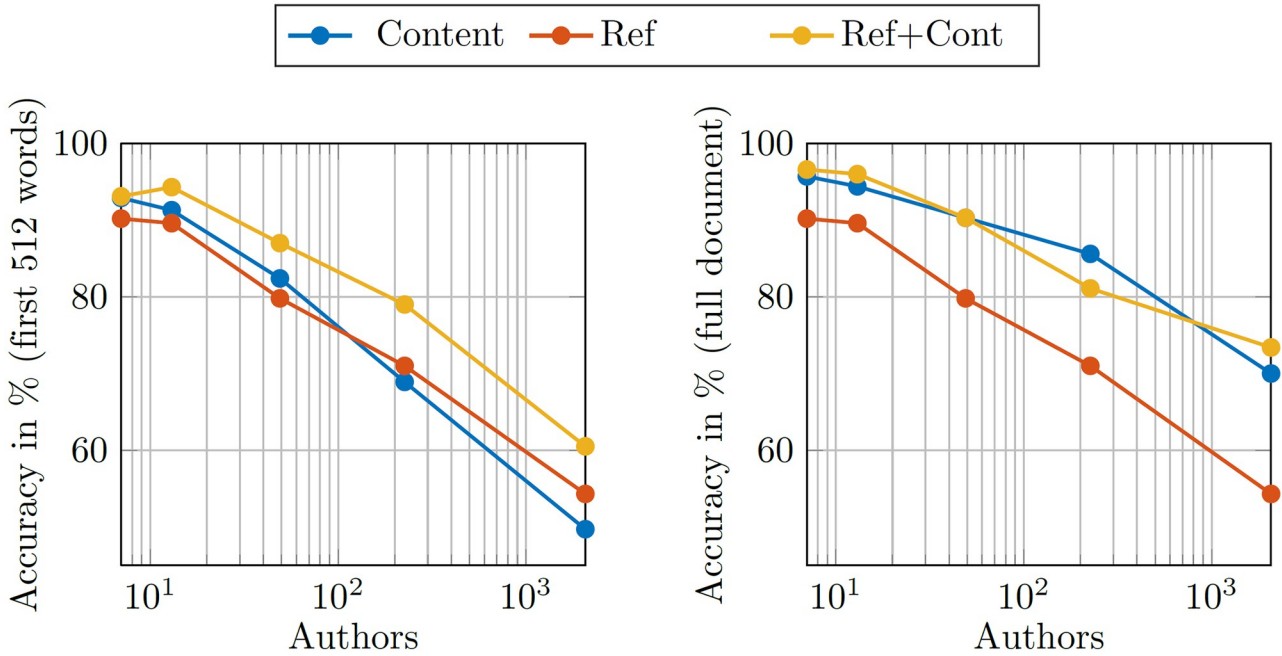

**Fig 3. The two plots visualize the results presented in Table 3.** On the left the 'non-C' datasets using only the first 512 words are used, on the right the full paper is used. Although the AA accuracy degrades with an increasing number of authors, our approach retains an impressive 73.4% for 2070 authors.

One can also argue that all cited references are somehow related to the content of the paper. In a case where the content network has access to the whole article, this might not add much new information. In the case where only the first 512 words are used, it is credible that the references add valuable information not included otherwise.

### 5.3 Scaling to larger datasets

The impressive performance of 73.4% correctly attributed papers in a pool of 2070 candidate authors gives rise to the question how large of a dataset the proposed approach can handle. Is it possible to just train on the entirety of arXiv and still obtain a reasonable performance with tens of thousands of candidate authors?

Unfortunately, due to computational limitations we are unable to train a network on such a large dataset as already 2070 authors with about 1.3 billion texts in total results in over one week of training time (see Table 4). However, in the following we will present a result that supports the assumption that, given enough compute to train a model, larger datasets with more

**Table 4. Comparison of training times on an Nvidia Quadro RTX 8000 GPU for the best model from Table 3. The last column reports the increase in accuracy when including the full document (C) is used.**

| Dataset | First 512 Words (non-C) | Full Document (C) | Δ Accuracy |
|---------|-------------------------|-------------------|------------|
| D100 | 13h:08m | 11d:04h:40m | 9.5% |
| D200 | 1h:35m | 17h:49m | 6.6% |
| D300 | 49m | 4h:35m | 3.3% |
| D400 | 17m | 1h:57m | 1.7% |
| D500 | 11m | 54m | 3.1% |

**Table 5. Results on the trimmed datasets with a decreasing number of papers per author available for training and testing.** Even at less than 1/8th of the original data, the network still retains 75% of its performance on the D200-C dataset. When the number of authors is increased 20 times with only 25 papers per author (D50T25-C), the performance drops by a mere 6 percentage points.

| Dataset | D200-C | | | | | D100T25-C | D50T25-C |
|---|---|---|---|---|---|---|---|
| Authors | 226 | 226 | 226 | 226 | 226 | 2070 | 5292 |
| Papers/Author | >200 | 200 | 100 | 50 | 25 | 25 | 25 |
| Accuracy [%] | 81.6 | 76.6 | 75.1 | 71.4 | 62.5 | 57.0 | 56.4 |

papers and authors will result in good performance. First, we vary the number of papers per author on the D200-C dataset (see Table 5). Training the model with less and less papers for each of the 226 authors degrades performance, but even with 1/8th of the papers per author the model retains 75% of its accuracy.

Thanks to the limited number of papers per author, we are able to increase the number of candidate authors past the D100 dataset and we generate a dataset which contains over 5000 authors and train our approach on it with 25 papers per author. Despite the D50T25-C having over 20 times more authors than the D200T25-C the accuracy only drops by 6 p.p. from 62.5% to 56.4%.

The two experiments show that while reducing the number of papers per author degrades the accuracy, it does not break our method. In fact the experiments show that our method scales well and including more authors has a surprisingly small effect on the overall accuracy. This is probably due to the transformer benefitting from the very large amounts of data that is available overall. It thus properly finetunes to the text modality 'research paper' (and the peculiarities of our data preparation pipeline) and still yields good results.

## 5.4 Analysis of the accuracy

In the previous sections we have always reported the average attribution accuracy and, in agreement with prior work, used this as a measure of the model performance. In this section we further analyze the attribution accuracy. The goal is to understand whether the model achieves a uniform attribution accuracy across the authors, or if it gets some authors nearly always right and others practically never. All of the following analysis is based on results obtained with our *Ref+Cont* architecture.

As a first step, we show the attribution accuracies for the D100-C, D200-C and D300-C datasets in Fig 4. The average attribution accuracies for each model are (from Table 3) 73.4%, 81.1%, and 90.3%, respectively. We see some variation in the attribution accuracy histogram across authors (Fig 4, left) and especially for the large D100-C dataset the model performs poorly for a few authors. At the same time, there are also many authors where the model gets close too 100% accuracy. The width of the attribution accuracy interquartile range is below 30 p.p. for all datasets as shown in Fig 4 (right). This proves that the model performs rather similar for most authors in the dataset.

Since the datasets are generated based on selecting authors that have *at least* a certain number of publications on arXiv, the number of training/testing samples per author varies within the dataset. In a second analysis step, we investigate this aspect in greater detail. Fig 5 shows a similar boxplot as Fig 4, but additionally considers the number of samples available per author (x-axis). The plot confirms what is intuitively true: authors with more available samples are easier to correctly identify. As such, large parts of the variability shown in the histogram in Fig 4 can be explained by the number of samples available per author.

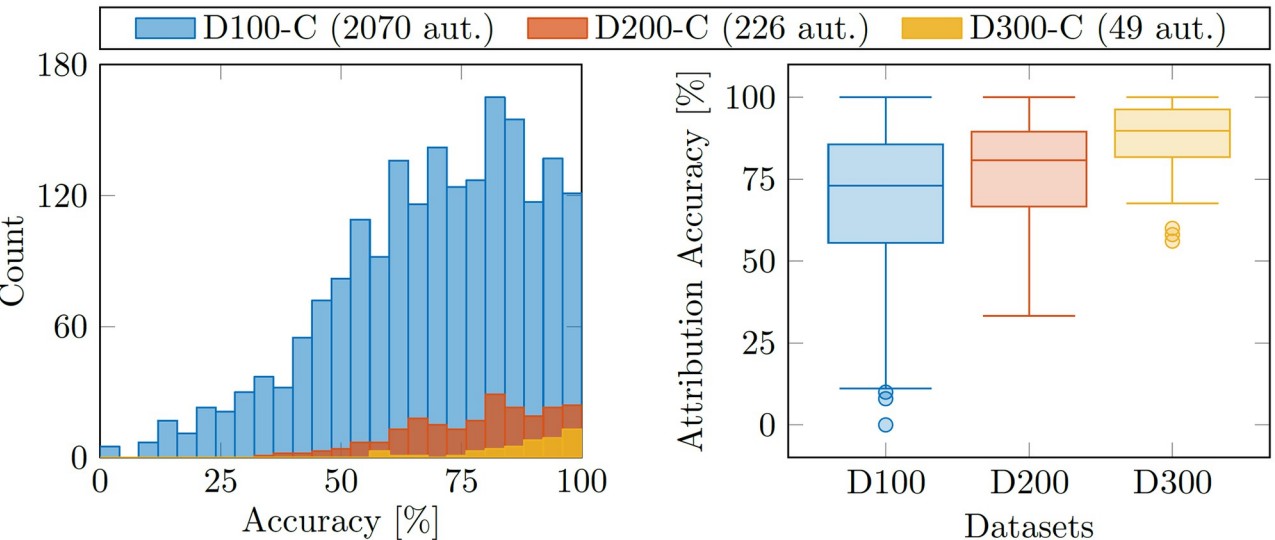

**Fig 4. On the left the attribution accuracy distribution for the three datasets D100-C, D200-C and D300-C is shown.** The boxplot on the right presents the summary statistics corresponding to the histogram on the left. Both diagrams show that our method performs relatively consistent across the authors in the dataset in terms of accuracy.

Looking at the accuracy as a function of the available paper samples in Fig 5 also reveals a more interesting detail: for small numbers of papers (in the range of 75 to 200 papers per author), the results obtained with the D100-C are superior to the D200-C. In other words, the 'harder' dataset with 2070 candidate authors achieves a higher attribution accuracy than the 'easier' D200-C dataset with only 226 authors. This supports the findings from the scaling analysis: the amount of data for training plays a crucial role, and our method performs very well in

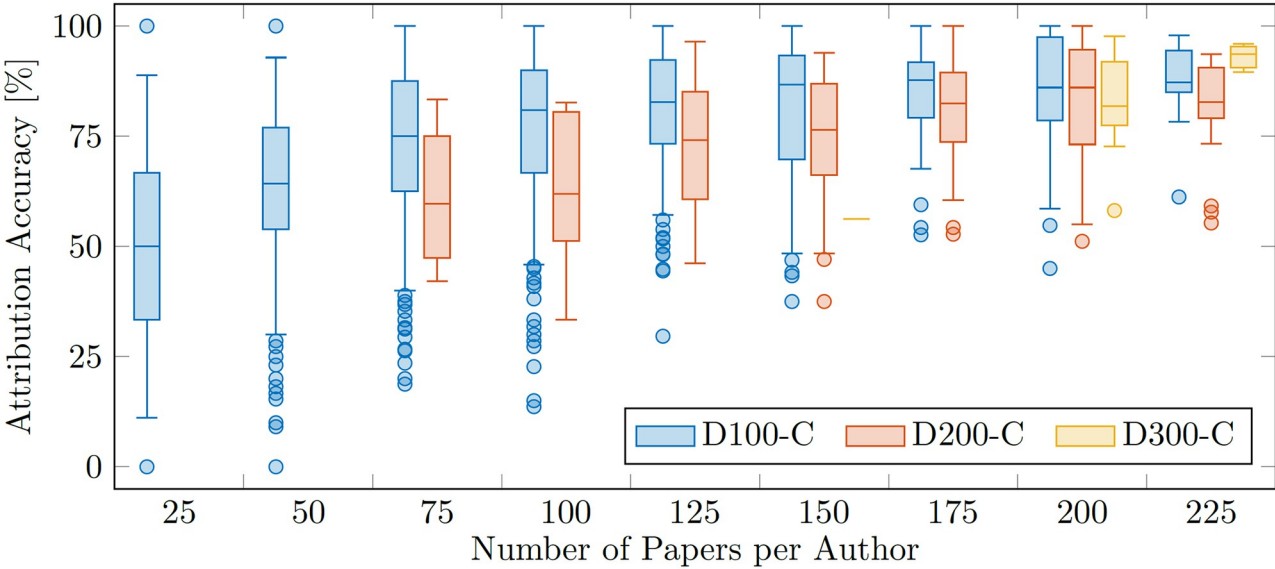

**Fig 5. The boxplot shows the attribution accuracy of our method as a function of the dataset size (color coding) and the number of papers per author (groups on the x-axis).** As intuitively expected, more samples per author increase the attribution accuracy. Interestingly, for a given number of papers per author (e.g. 100) one can see that an increased overall dataset size (e.g. D100-C vs. D200-C) yields higher attribution accuracy.

presence of many candidate authors if sufficient amounts of training data are given. Interestingly, Fig 5 shows that in this case even training data for *more* authors is helpful and increases the overall accuracy. Thus, given the compute required to actually train our method, it is expected to scale well to even larger datasets.

## 5.5 Multi-author predictions

Although our approach was trained to predict only one author, we study the multi-author prediction capabilities of our model to better understand the limitations. For this study we use the D100-C dataset. The test split contains 4% papers with at least two authors being in the set of the 2070 candidate authors. For the majority only one author is part of the candidate set.

For clarity we refine the notation first. The set of the 2070 candidate authors is denoted with $\mathcal{C}$. The (groundtruth) set of co-authors that wrote a paper is denoted with $\mathcal{G}$. It holds that $\mathcal{G} \subset \mathcal{C}$. Our model predicts logits that represent the odds of a certain author being a co-author of a paper. The set $\mathcal{M}_{1:N}$ denotes the $N$ most likely authors, e.g. $\mathcal{M}_1$ is the most likely author, $\mathcal{M}_{1:3}$ are the three most likely authors. We analyze the attribution accuracies for the following four evaluation metrics:

1. The most likely author identified by our model is *an* author of the paper. This is the metric we have studied throughout our study so far. Formally, $\mathcal{M}_1 \subseteq \mathcal{G}$.

2. Given the number of authors ($n = |G|$), *all* the authors of the paper are identified by our model. Formally, $\mathcal{M}_{1:n} = \mathcal{G}$.

3. Estimating the number of authors $\hat{n} = f(\mathcal{M})$ using some function *f*, *all* the authors of the paper are identified by our model. Formally, $\mathcal{M}_{1:\hat{n}} = \mathcal{G}$.

4. All the authors are part of the top $k = 5$ predictions. Formally, $\mathcal{G} \subseteq \mathcal{M}_{1:k}$.

For metric (3) we require a function *f* that decides, based on the model's prediction, which authors are likely still a co-author. We chose a simple function which considers the predictions as authors if their probability is at least 0.1 times the most probable prediction. This threshold has been tuned empirically to yield good performance on the D100-C dataset.

Table 6 shows the results for the metrics (1)-(4) introduced above. Note that the 'single author' or 'multi authors' criterion does not say anything about whether the paper was written by multiple reseachers, it only refers to multiple authors being part of the dataset. *Multi author* papers thus have more than two co-authors that are included in the set of candidate authors, whereas *single author* papers have only one coauthor who is part of the dataset. For the first metric we observe that papers with multiple authors achieve a higher accuracy. Intuitively, this makes sense as it is more likely that the predicted author $\mathcal{M}_1$ is an author, if a paper has multiple co-authors. For a similar reason, single author papers achieve a higher score in metric (4) as it is more likely that the one author is included in the set of the predicted five most likely authors.

**Table 6. Accuracy [%] for the metrics (1)-(4).** For a more finegrained analysis we present the results in the entire dataset (overall) as well as when only the papers with one (single author) or many authors (multiple authors) are selected for evaluation.

| | Metric (1) $\mathcal{M}_1 \subseteq \mathcal{G}$ | Metric (2) $\mathcal{M}_{1:n} = \mathcal{G}$ | Metric (3) $\mathcal{M}_{1:\hat{n}} = \mathcal{G}$ | Metric (4) $\mathcal{G} \subseteq \mathcal{M}_{1:5}$ |
|---|---|---|---|---|
| Single Author | 73.2 | 73.2 | 62.2 | 88.2 |
| Multi Author | 79.3 | 47.1 | 17.8 | 67.9 |
| Overall | 73.4 | 72.3 | 60.7 | 87.6 |

In the metric (2), where the number of authors is known, our approach still achieves nearly 50% correct identification of *all* coauthors for multi-author papers. In the metric (3), where the number of authors is unknown, the accuracy drops to about 20% for multi-author papers. Additionally, the accuracy for single-author paper drops, i.e., the model assigns two or more authors to a single-author paper.

We conclude that our approach performs very well on multi-author predictions (metric 1). If the task is to correctly predict *all* of the authors, our model only performs well, given that the number of authors is known (metric 2). If the exact number of authors is unknown but one is only interested in a set of 5 candidate authors that are likely authors (metric 4), in over 2/3 of the metrics all candidate authors are found among the top-five suggestions of the model.

In the context of plagiarism detection, metric (4) is especially relevant as it allows further checking the relevant authors with a more targeted but less efficient method. Metric (1) is interesting if the goal is to crack a double-blind review process as knowing one of the authors is usually enough to know from which group or lab a work comes.

## 5.6 Ablations

**5.6.1 Learning rate.** In order to obtain the final results that are reported in Table 3, a fine-tuning stage of the learning rate was needed. The evaluation accuracy for different learning rates for some of our datasets is shown in Table 7. The learning rates of the rows in bold are selected for all runs of that type, e.g. all Ref+Cont architectures are trained with a learning rate of 5e-5 for whole documents.

**5.6.2 Alternative architecture.** In this section we present the results for the alternative LSTM architecture to encode the references. To evaluate the accuracy of the FastText embedding and LSTM model, we train this network to predict the authors using only the references. This achieved an accuracy of 75.03% on the D300 dataset, which is slightly worse than RHE model. When combined with the DistilBERT, the accuracy increases to 81.4% on *D*300 but falls short of the RHE baseline for Ref+Context. This indicates that LSTM is not a suitable architecture for this task. Intuitively this makes sense as sequential information of the author names in the references is not significant, but rather the frequency of author names is useful for predicting author attribution.

We also evaluated the performance of FastText embedding in combination with 2 layer MLP by averaging the embedding for all the authors corresponding to the references for each paper. However, this approach too did not perform any better that RHE. Additionally, we also

**Table 7. Ablation of the learning rate for 10 epochs.**

| Rate | Content | References | | Ref+Cont | |
|---|---|---|---|---|---|
| | D300 | D300 | D500 | D300 | D500-C |
| 1e-5 | 78.8 | | | | 93.1 |
| 2e-5 | 81.3 | | | | 95.7 |
| 5e-5 | 81.3 | | | | **96.7** |
| 1e-4 | **82.4** | 79.3 | 86.7 | 84.5 | 95.7 |
| 2e-4 | 82.1 | **80.2** | 87.6 | 86.5 | 94.1 |
| 3e-4 | | | | **87.4** | |
| 4e-4 | 81.2 | 79.6 | 89.2 | 87.2 | |
| 8e-4 | | 79.4 | **90.4** | 85.4 | |
| 2e-3 | | 80.0 | 90.3 | | |

tried to use another DistilBERT in parallel only for the references. The hypothesis was that the transformer would learn the underlying structure of the references and that it would be able to learn extract the key information. However, the final classification accuracy was lower and the training times were, at least, slowed down by a factor of 2. Therefore, these architectures were discarded.

## 6 Conclusion & discussion

We presented a transformer-based classification architecture for research papers that leverages, for the first time, a combination of the syntactic richness and topic diversity contained in research content and the information contained in the reference section. Our results show that combining both sources of information increases the authorship attribution accuracy. In cases where only limited text content is available to the network, including references increases the performance significantly (up to 11%). Overall, our method achieves 73.4% accuracy on the D100-C dataset, containing over 2000 authors, which is unprecedented to the best of our knowledge. In order to conduct this work, we also present a large-scale authorship-identification dataset by leveraging 2 million research papers publicly available on arXiv. On smaller datasets ($<$50 authors) and some benchmarks, the proposed architecture robustly identifies an author correctly well over 90% of the time, beating state-of-the-art results.

While our DistilBERT-based approach outperforms all baselines on large datasets such as *Legal* and *IMDb62*, it fails to outperform simple n-gram baseline on small datasets. The explanation here is that the data-hungry nature of transformers limits the performance of our approach on smaller datasets. When studying larger datasets, we have shown in our scaling analysis that the method can be expected to scale very well given the computational capabilities to train it. This is crucial for research paper authorship attribution which deals with very large datasets.

Analyzing the accuracy of our method in detail we find that it performs similar across most authors in the dataset, although authors with more available samples are easier to identify. Additionally, we study the attribution accuracy if multiple authors are to be identified. If the task is to correctly predict all of the authors, our model performs well and still achieves close to 50% accuracy, given that the number of authors is known.

If the exact number of authors is unknown, but one is only interested in a set of 5 candidate authors that could be authors, in 67% of the cases all candidate authors are among the top-five suggestions of the model. When the approach is used in the real-world to support plagiarism detection, this last case is especially relevant as it allows further checking of the relevant authors with a more targeted (but possibly less efficient) method.

We believe that this line of research—albeit having great implications for double-blind review—ultimately helps to improve the review process. Thus, we conclude our paper by summarizing the key insights into how a submission can remain anonymous in order to support an unbiased, double-blind review process.

- *Abstract and introduction*: already the first 512 words enable robust authorship attribution. We believe that this is because the abstract and introduction often express the authors creative identity together with the research field. These personalized characteristics enable identification of the authors.

- *Self-citations*: The papers in our dataset contain, on average, 10.8% self-citations. Those citations easily give away the authors' identity as highlighted by the results shown in Table 3. Therefore, it is beneficial to omit many self-citations in the submission for double-blind review.

- *Citation diversity*: Even without the self-citations, the references can be used to identify the author. By also including citations of less well-known papers authors can make authorship attribution more difficult. At the same time, more equal visibility is given to all research papers in the authors' field.

Based on the experiments we conducted, we are the first to offer insights that validate some hypotheses. A crucial finding in our research is the ability to predict the authors of a paper using publicly available large scale data. This poses a direct threat to the integrity of the double blind review process. To mitigate this issue, we suggest simple actions such as reducing self-citations, which can be implemented during the desk rejection stage.

We hope that this study encourages the community to investigate authorship attribution for research papers further as possible applications like plagiarism detection and improving the double-blind review process are relevant to the entire scientific community.

## 7 Ethical considerations

The task of AA for research papers has some ethical concerns as it offers a potential way of breaking the double blinded peer review system, a pillar of academic research. While the proposed methodology challenges this double blind peer review system by uncovering the author identity only from text and references, we believe our method can help establishing an improved peer review system. By analysing our method and providing insights into how a paper can be attributed to an author, we hope to guide authors towards a writing style that improves double-blind review. Therefore, we believe that the possible negative consequences are outweighed by the opportunity of this exciting research direction, which has not been thoroughly pursued in the past. For this very reason, we also open-source all the tools required to reproduce our experiments and further develop our methodology under https://github.com/uzh-rpg/authorship_attribution.

## Acknowledgments

**Code**: https://github.com/uzh-rpg/authorship_attribution.

## Author Contributions

**Data curation:** Leonard Bauersfeld.

**Funding acquisition:** Davide Scaramuzza.

**Methodology:** Leonard Bauersfeld, Angel Romero, Manasi Muglikar.

**Project administration:** Leonard Bauersfeld.

**Software:** Leonard Bauersfeld, Angel Romero, Manasi Muglikar.

**Supervision:** Davide Scaramuzza.

**Writing – original draft:** Leonard Bauersfeld, Angel Romero, Manasi Muglikar, Davide Scaramuzza.

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
