## [Decision Letter · Decision Letter 0]

12 Mar 2023

PONE-D-23-00696Cracking double-blind review: authorship attribution with deep learningPLOS ONE

Dear Dr. Bauersfeld,

Thank you for submitting your manuscript to PLOS ONE. After careful consideration, we feel that it has merit but does not fully meet PLOS ONE’s publication criteria as it currently stands. Therefore, we invite you to submit a revised version of the manuscript that addresses the points raised during the review process.

 Both reviewers agree the paper has margin to improve, specifically regarding the following points:1) The behavior of the proposed system when trained on a dataset where each author has fewer than 100 papers.

2) A thorough discussion on the results, in terms of false attribution, correlations, pitfalls, limitations when used in a real-world scenario, etc.If the authors could invest their time on those two particular points, I believe the resulting paper will be much stronger and useful overall.

We look forward to receiving your revised manuscript.

Kind regards,

Rodrigo Coelho Barros, Ph.D.

Academic Editor

PLOS ONE

Journal Requirements:

Journal Requirements:

Important: If there are ethical or legal restrictions to sharing your data publicly, please explain these restrictions in detail. Please see our guidelines for more information on what we consider unacceptable restrictions to publicly sharing data: http://journals.plos.org/plosone/s/data-availability#loc-unacceptable-data-access-restrictions  Note that it is not acceptable for the authors to be the sole named individuals responsible for ensuring data access.

Reviewers' comments:

Reviewer's Responses to Questions

**Comments to the Author**

1. Is the manuscript technically sound, and do the data support the conclusions?

Reviewer #1: Yes

Reviewer #2: Partly

2. Has the statistical analysis been performed appropriately and rigorously? 

Reviewer #1: Yes

Reviewer #2: N/A

3. Have the authors made all data underlying the findings in their manuscript fully available?

Reviewer #1: Yes

Reviewer #2: Yes

4. Is the manuscript presented in an intelligible fashion and written in standard English?

Reviewer #1: Yes

Reviewer #2: Yes

5. Review Comments to the Author

Reviewer #1: The manuscript introduces the interesting and significant task of author attribution to the large arXiv pre-print service. They explain their methodology and results very clearly, and I am confident their results are correct as presented in the main text. My first main suggestion is to make consistent the claims in the abstract with the findings of the paper. The abstract implies that the D100 data set (with about 2000 authors) yields up to 95% correct attribution (when the text reports up to 70%). I understand the operative use of 'up to' makes the claim in the abstract technically true, but it appeared misleading when I got the the presented results. My second main criticism is that the data set choices are limited in a strange way. Why did the authors structure their datasets with so few authors? I understand they choose to segment the datasets by authors with a certain number of papers, which is a fine choice, but why at the most consider authors with at least 100 papers? The choice of even considering D500 which has only a tiny number of authors also seems strange, because of what practical value is choosing among 7 potential authors? This is by far the largest technical weakness of the paper in my opinion, and it very much limits the significance of the findings. The authors need to address this issue, ideally by adding an experiment with almost all authors, or explaining technical limitations which preclude this possibility and suggesting steps to overcome them.

My third main suggestion is that the paper needs a more significant discussion of false attributions, which if this tool is used in some capacity for decision making or plagiarism detection etc, is a critical risk to assess. For example, they report how often they can identify a single author of a paper, but if they try to identify all authors, how many do they get right, and how many are falsely attributed?

Reviewer #2: The paper proposes a multichannel NN to predict authorship. It is based on a destilbert representation of paper content and a histogram-based representation of cited authors. To train the network, the authors created a public dataset from arxiv. The results are reproduceable, as the code is publicly available, including the code to create the dataset.

The application is interesting, and it constitute in the academic world a real issue. Authors, intentionally or unintentionally, provide many hints about authorship in double-blind submissions. In that regard, the motivation is nice, but the paper was very superficial in regards to the proposition and the experiments.

a) The paper is brief and provides few details about the proposed method. One needs to dig into the code to understand how the text was processed, the references were pre-pared, the details of the NN illustrated in Fig. 2, etc. The availability of the code is very important, but the paper need to be self-contained. There are many choices that need to be further justified. This is necessary in the dataset preparation, and on the NN architecture.

b) The authors chose to train the NN using people with at least 100 papers. While many faculties in the academic career may display such behavior, this is not true for younger researchers, say PH.D Candidates. I miss a discussion on that choice, and on the behavior of the system regarding authors with fewer papers.

c) I miss a discussion about the correct and incorrect predictions. I wonder whether only very well “known” authors are recognized, and what features lead to successful/unsuccessful predictions.

d) The authors claim in the introduction that they provide insights on how to identify authorship, but these insights are somewhat obvious, except perhaps for some “style” of writing introduction/abstract. Again, a more detailed discussion on that needs to be provided.

6. PLOS authors have the option to publish the peer review history of their article (what does this mean?). If published, this will include your full peer review and any attached files.

Reviewer #1: No

Reviewer #2: No

---

## [Author Response · Author response to Decision Letter 0]

28 Apr 2023

Please see our "Response to the Reviewers" file for our response to the comments made by the editor and the reviewers.

---

## [Decision Letter · Decision Letter 1]

8 Jun 2023

Cracking double-blind review: authorship attribution with deep learning

PONE-D-23-00696R1

Dear Dr. Bauersfeld,

We’re pleased to inform you that your manuscript has been judged scientifically suitable for publication and will be formally accepted for publication once it meets all outstanding technical requirements.

Kind regards,

Rodrigo Coelho Barros, Ph.D.

Academic Editor

PLOS ONE

Additional Editor Comments (optional):

The authors seem to have addressed all major points raised by the reviewers.

Reviewers' comments:

Reviewer's Responses to Questions

**Comments to the Author**

1. If the authors have adequately addressed your comments raised in a previous round of review and you feel that this manuscript is now acceptable for publication, you may indicate that here to bypass the “Comments to the Author” section, enter your conflict of interest statement in the “Confidential to Editor” section, and submit your "Accept" recommendation.

Reviewer #2: All comments have been addressed

2. Is the manuscript technically sound, and do the data support the conclusions?

Reviewer #2: Yes

3. Has the statistical analysis been performed appropriately and rigorously? 

Reviewer #2: Yes

4. Have the authors made all data underlying the findings in their manuscript fully available?

Reviewer #2: Yes

5. Is the manuscript presented in an intelligible fashion and written in standard English?

Reviewer #2: Yes

6. Review Comments to the Author

Reviewer #2: I think the authors have significantly improved the manuscript, and the analysis performed, increasing the novelty and usefulness of the proposed approach. I'd suggest two minor improvements, in this or future work of the authors:

- restricted use of superlatives such as "impressive". I do agree the authors have achieved good performance, but maybe it is not thtat "impressive".

- for clarity, the pre-processing text could be presented as a pseudo-algorithm.

7. PLOS authors have the option to publish the peer review history of their article (what does this mean?). If published, this will include your full peer review and any attached files.

Reviewer #2: **Yes: **KARIN BECKER

---

## [Editor Report · Acceptance letter]

20 Jun 2023

PONE-D-23-00696R1 

Cracking double-blind review: authorship attribution with deep learning 

Dear Dr. Bauersfeld:

I'm pleased to inform you that your manuscript has been deemed suitable for publication in PLOS ONE. Congratulations! Your manuscript is now with our production department. 

Kind regards, 

on behalf of

Dr. Rodrigo Coelho Barros 

Academic Editor

PLOS ONE